# Effect of Grape Pomace Intake on the Rumen Bacterial Community of Sheep

**Michal Rolinec** [1,*], **Juraj Medo** [2], **Michal Gábor** [1], **Martina Miluchová** [1], **Milan Šimko** [1], **Branislav Gálik** [1], **Ondrej Hanušovský** [1], **Zuzana Schubertová** [3], **Daniel Bíro** [1], **Luboš Zábranský** [4] **and Miroslav Juráček** [1,*]

[1] Institute of Nutrition and Genomics, Slovak University of Agriculture in Nitra, Trieda A. Hlinku 2, 94976 Nitra, Slovakia

[2] Institute of Biotechnology, Slovak University of Agriculture in Nitra, Trieda A. Hlinku 2, 94976 Nitra, Slovakia

[3] Institute of Plant and Environmental Sciences, Slovak University of Agriculture in Nitra, Trieda A. Hlinku 2, 94976 Nitra, Slovakia

[4] FZT Department of Animal Husbandry Sciences, University of South Bohemia, Branišovská 1645/31a, 37005 České Budějovice, Czech Republic

[*] Correspondence: michal.rolinec@uniag.sk (M.R.); miroslav.juracek@uniag.sk (M.J.)

**Abstract:** The performance of ruminants is affected mainly by the rumen bacterial community. The composition and properties of the rumen bacterial community depend largely on the diet components that are fed to the ruminant. The aim of this study was to determine the effect of grape pomace intake on the rumen bacterial community of sheep. Four different diets, two of which contained dried grape pomace (DGP), were used in this study. Rumen fluid samples from 12 wethers were used for 16S rRNA gene sequencing and subsequent bacterial identification. At the phylum level, Bacteroidetes and Firmicutes; at the family level, Prevotellaceae and Porphyromonadaeceae; and at the genera level, *Prevotella* and *Verrucomicrobia_Subdivision5_genera_incertae_sedis* were the most common regardless of the diet the animals were fed. After the addition of DGP to the diet, the relative abundance of *Methanobrevibacter*, *Butytirivibrio*, *Fretibacterium*, and *Verrucomicrobia _Subdivision3 _genera_incertae_sedis* significantly increased, whereas that of *Succiniclasticum* and *Selenomonas* significantly decreased. The upregulated pathway of methanogenesis from $H_2$ and $CO_2$ was supported by a significant increase in *Methanobacteriaceae* after the diet was supplemented with DGP. The rumen bacterial community diversity indices (Richness and Shannon) were significantly affected by diet composition as well as by the change of housing location. The addition of DGP into the wethers' diet increased the richness of the rumen bacterial community, which is good for maintaining rumen bacterial homeostasis. No adverse effect of the addition of DPG on the rumen bacterial community was noted.

**Keywords:** wine industry by-products; grape pomace; sheep; rumen bacteria; abundance; diversity

## 1. Introduction

The essential presumption of the full utilization of ruminants' production potential is the maintenance of a stable rumen environment as well as high activity of rumen bacteria. From a nutritional point of view, the quantity and quality of ruminant performance is conditioned by feeds, which affects the production of volatile fatty acids and microbial protein in the rumen [1,2]. For this, a stable supply of nutrients to the rumen is required. Every change in ruminants' diets causes alterations in the rumen bacterial population. If there is a negative alteration, the efficiency of production is also negatively affected [3].

Findings published by Clark [4] indicated that nutrition is a key factor affecting the rumen bacterial community. The utilization of by-products for feeding animals is preferred nowadays. Around the world, approximately 70 million tons of grapes are produced annually [5], which creates a huge amount of grape pomace. As stated in [6], grape by-products have potential applications in animal nutrition as a source of bioactive compounds. Grape pomace is a source of polyphenols, including tannins, which cause

complexation between the tannin and protein, thus positively affecting the amount of dietary ruminal undegraded protein [7,8]. Grape pomace as part of a ruminant diet has been researched by many authors [9–11]. Additionally, the effect of grape pomace on the rumen microbiota has been researched. Vasta et al. [12] found that polyphenols can violate the rumen microbiota composition. On the other hand, Salami et al. [13] concluded that polyphenol-rich feed (cardoon meal) can be used for ruminants without negatively affecting the rumen microbiota. Biscarini et al. [14] found an increase in the diversity of rumen microbiota after feeding grape pomace as a source of natural polyphenols.

Research on the dietary effects on the rumen microbiota is highly topical, and the results of experiments with grape pomace and the gut bacterial community are not uniform; this study was designed to address some of these issues. The effects of dietary change, together with the shift of the location and the effect of the addition of grape pomace on the bacterial community of the rumen, were investigated.

## 2. Materials and Methods

### 2.1. Animals and Experiment Design

In this experiment, 12 wethers (Ile de France breed) with an average weight of $35.28 \pm 2.04$ kg were included. For bacterial community analyses, wethers were marked as individual A, B, C, ... L. Wethers were housed in the University Farm (Slovak University of Agriculture in Nitra, Nitra, Slovakia) in Žirany before the experiment. The experiment was conducted at the barn for farm animals in Nitra (Slovak University of Agriculture in Nitra, Nitra, Slovakia). The feeding regime and rumen fluid collection procedure are shown in Figure 1. In each period of the experiment, the same twelve wethers were used. This provided the same effect because the animals' genetic makeup, condition, and health status were the same for each experimental period. First, rumen fluid samples were collected immediately after the transport of wethers from University Farm in Žirany to the barn for farm animals in Nitra (Slovak University of Agriculture in Nitra, Nitra, Slovakia), where the experiment was carried out (Timepoint 1). After collection of the first sample, a 19-day preparatory period followed, feeding the wethers a control diet. After the end of the preparatory period, the second rumen fluid samples were collected (Timepoint 2). Subsequently, wethers were fed a control diet containing 1% DGP for 12 days. On the last day of the first experimental period, the third rumen fluid sample was collected (Timepoint 3). Then, the second experimental period started, where animals were fed with a control diet containing 2% DGP for 12 days. During the last day of the second experimental period, the fourth rumen fluid sample was collected (Timepoint 4) (Figure 1). The compositions of all diets are shown in Table 1. For all diets, mineral and vitamin licks and water were available ad libitum throughout the entire study period. DGP was obtained after the drying of pomace after grape (Pinot Gris variety) pressing. The pomace obtained after grape pressing consisted of skin, pulp, and seeds. Drying was conducted for 4 days at $55 \pm 5$ °C. The grape pomace was acquired from the Academic Winery of the Slovak University of Agriculture in Nitra, located in the village of Oponice (Slovakia). To ensure that the only feed intake that occurred was from the tested diets, from the start of the preparatory period to the end of the second experimental period, wethers were housed in a pen without any bedding. Wethers were kept under standard conditions and cared for by trained and experienced staff under the care of the contracted veterinary doctor. The protection standards for animals used for scientific and teaching purposes were in accordance with national directive no. 2010/63/EU [15].

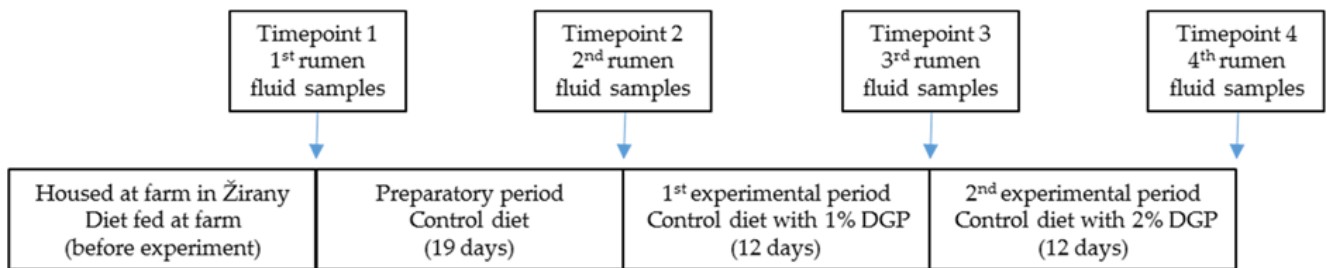

**Figure 1.** Timepoints of rumen fluid sampling.

**Table 1.** Diet compositions before and during the experiment.

| | Diet Fed at Farm in Žirany | Control Diet | Control Diet with 1% DGP | Control Diet with 2% DGP |
|---|---|---|---|---|
| Feeds (g) | | | | |
| Maize silage | 400 | - | - | - |
| Alfalfa silage | 300 | - | - | - |
| Barley grounded | 100 | - | - | - |
| Meadow hay | 500 | 700 | 700 | 700 |
| Wheat grounded | 100 | 118.6 | 118.6 | 118.6 |
| Soybean meal | - | 238.6 | 238.6 | 238.6 |
| DGP | - | - | 10.3 * | 20.6 ** |
| Nutrient concentrations in diet (g) | | | | |
| Dry matter | 838.4 | 934.1 | 943.8 | 953.5 |
| Crude protein | 88.1 | 179.0 | 180.0 | 181.0 |
| Ether extract | 16.4 | 13.0 | 13.9 | 14.8 |
| Crude fiber | 265.8 | 288.0 | 289.9 | 291.8 |
| ADF | 310.5 | 351.4 | 355.3 | 359.2 |
| NDF | 474.8 | 529.8 | 534.5 | 539.3 |
| NFE | 491.2 | 520.6 | 526.8 | 532.9 |
| NFC | 282.2 | 278.8 | 282.1 | 285.4 |
| Organic matter | 861.4 | 1000.6 | 1010.5 | 1020.4 |
| Ash | 49.0 | 56.6 | 57.0 | 57.4 |
| Ca | 4.2 | 4.1 | 4.1 | 4.2 |
| P | 2.4 | 3.9 | 4.0 | 4.0 |
| Mg | 1.5 | 2.1 | 2.1 | 2.1 |
| Na | 0.3 | 0.3 | 0.3 | 0.3 |
| K | 11.1 | 15.5 | 15.6 | 15.8 |

* indicates 1% from daily dry matter intake; ** indicates 2% from daily dry matter intake; DGP, dried grape pomace; ADF, acid detergent fiber; NDF, neutral detergent fiber; NFE, nitrogen-free extract; NFC, non-fiber carbohydrates.

*2.2. Diet Composition and Analysis*

The nutritional characteristics of the diets fed to wethers are shown in Table 1. Diets were analyzed in the Laboratory of Quality and Nutritional Value of Feeds at the Institute of Nutrition and Genomics (Slovak University of Agriculture in Nitra), according to official AOAC laboratory methods [16].

*2.3. Rumen Fluid Sample Collection*

Rumen fluid was collected on the morning of the last day of each period, before feeding. After the wether had been secured, 2 mL rumen fluid was collected using duodenal sound type Levin CH14 (Unomedical, ConvaTec, Greenboro, NC, USA). Sampled rumen fluid was mixed with 2 mL of DNA/RNA preservation reagent (DNA/RNA Shield™, Zymoresearch, Irvine, CA, USA). Rumen fluid samples were stored at −80 °C.

*2.4. DNA Analysis*

Bacterial DNA extraction was performed using the EZ10 bacterial extraction kit (BioBasic, Markham, ON, Canada). For assessment of the bacterial community, the V4 part of the 16S gene was amplified using barcoded primers 515F and 806R [17]. The reaction mixture (25 μL) contained 1U of Q5 DNA polymerase (New England Biolabs, Inc., Ipswich, MA, USA), 12.5 μL Q5 PCR buffer (New England Biolabs, Inc., Ipswich, MA, USA), 5 μL of each primer (2 μM) (Generi Biotech, Hradec Králové, Czech Republic), and 2 μL of DNA. The thermal cycling conditions were as follows: initial denaturation 98 °C for 3 min; 30 cycles of 98 °C denaturation for 3 s; 65 °C annealing for 10 s; and amplification at 72 °C for 20 s. PCR products were checked on an agarose gel, purified by Ampure XP (Beckman Coulter, Brea, CA, USA), quantified by qubit (Invitrogen, Carlsbad, CA, USA), and pooled in equimolar ratios. Sequencing libraries were prepared using the Illumina TruSeq PCR free library prep kit (Illumina, San Diego, CA, USA). Sequencing was performed on Illumina MiSeq using a V3 2x300 bp kit. Acquired sequences were initially processed using SEED 2.1 software [18], where sequences were demultiplexed according to their barcodes and quality-filtered. Then, the DADA2 algorithm in QIIME 2 [19] was used for sequence denoising and amplicon sequence variant (ASV) calling. ASVs were identified using the Ribosomal Database Project (RDP) Classifier ver. 2.13. [20]. Indexes of diversity (richness, Pielou evenness, and Shannon's diversity index) and a Unifrac similarity matrix were calculated using QIIME 2. Picrust2 was used for the prediction of active metabolic pathways annotated by the MetaCYC database.

*2.5. Statistical Analysis*

Statistical analysis was performed in R [21]. The abundances of particular ASVs, genera, families, or phyla were analyzed using DeSeq2 [22]. Diversity indices among timepoints were compared using paired *t*-tests. Similarities in community composition among samples were visualized using NMDS and statistically analyzed by PERMANOVA using a two-factor model involving variability in timepoints and between individuals. PERMANOVA, pairwise comparisons, and betadisper homogeneity of dispersion was analyzed using the vegan package [23]. Metabolic pathways predicted in PICRUSt2 [24] were statistically compared between timepoints using DeSeq2, and volcano plots were created using ggplot in R.

**3. Results**

In total, 2236 ASVs were identified in rumen fluid, regardless of the diet that was fed and irrespective of the sampling timepoint. When the distribution of ASVs at each timepoint was considered, the high proportion of ASVs was specific for timepoint 1 and timepoint 4. Only 6.2% of ASVs were common across all four sampling timepoints (Figure 2).

The Bacteroidetes, Firmicutes, and Verrucomicrobia phyla were the most abundant in rumen fluid. These were followed by Proteobacteria, Euryarchaeota, Synergistetes, SR1, and Candidatus Saccharibacteria. The phyla with an abundance of at least 1% in at least one sampling timepoint were Actinobacteria, Fibrobacteres, Spirochaetes, Chloroflexi, and Tenericutes. The change of housing together with the change from the diet fed at the University Farm in Žirany to the control diet (TP1 to TP2) caused a shift in the relative abundance of some bacteria, which was significant for 9 of the 13 most abundant phyla. On the other hand, the addition of 1% DGP to the control diet (TP2 to TP3) caused only one significant shift by Synergistetes ($p < 0.001$). An increase in DGP in the diet from 1% to 2% (TP3 to TP4) caused four significant changes (Euryarchaeota, Candidatus Saccharibacteria, Fibrobacteres, and Verrucomicrobia). In comparison with the control diet, in the control diet with the addition of 2% DGP (TP2 to TP4), the abundance of Euryarchaeota and Candidatus Saccharibacterium increased, whereas that of Spirochaetes, Tenericutes, and Verrucomicrobia decreased (Table 2).

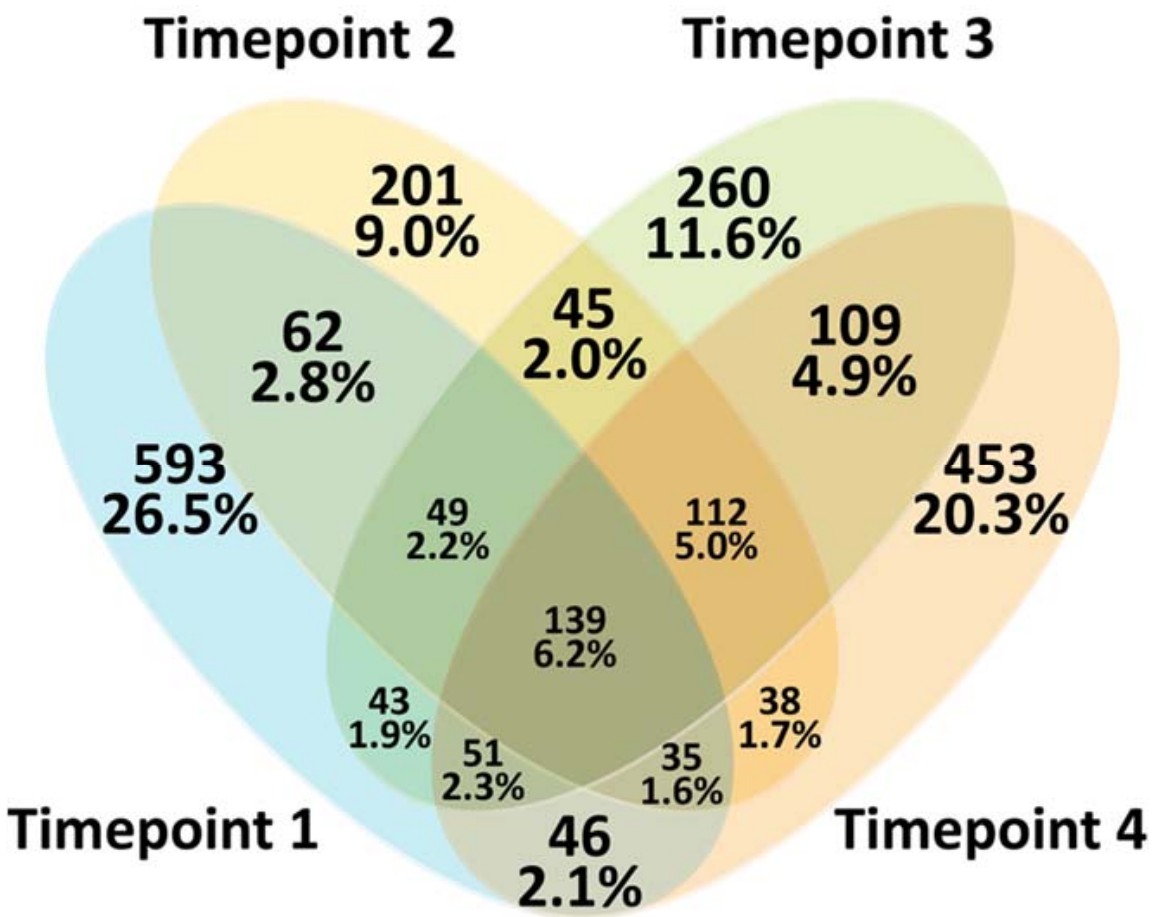

**Figure 2.** Venn diagram of amplicon sequence variants (ASVs) exhibited at different rumen fluid sampling timepoints.

**Table 2.** Dominant phyla with a relative abundance of over 1.0% at one or more of the sampling timepoints.

| Phylum (%) | TP1 | TP2 | TP3 | TP4 | *p*-Value TP1 to TP2 | *p*-Value TP2 to TP3 | *p*-Value TP3 to TP4 | *p*-Value TP2 to TP4 |
|---|---|---|---|---|---|---|---|---|
| Actinobacteria | 2.50 | 0.15 | 0.03 | 0.29 | **<0.001** | NA | NA | 0.291 |
| Bacteroidetes | 47.99 | 61.35 | 58.35 | 56.07 | 0.946 | 0.953 | 0.838 | 0.561 |
| Candidatus Saccharibacteria | 2.98 | 0.30 | 0.34 | 1.97 | **<0.001** | 0.893 | **0.006** | **0.001** |
| Chloroflexi | 0.30 | 1.28 | 0.87 | 1.30 | 0.052 | 0.632 | 0.996 | 0.395 |
| Euryarchaeota | 1.34 | 1.30 | 2.20 | 7.74 | 0.946 | 0.976 | **<0.001** | **<0.001** |
| Fibrobacteres | 1.64 | 0.46 | 0.64 | 0.25 | **<0.001** | 0.976 | **0.006** | 0.261 |
| Firmicutes | 23.85 | 20.15 | 15.59 | 16.92 | **0.005** | 0.428 | 0.644 | 0.056 |
| Proteobacteria | 3.59 | 1.56 | 3.80 | 3.83 | **0.001** | 0.368 | 0.514 | 0.516 |
| Spirochaetes | 1.59 | 0.82 | 0.08 | 0.11 | **0.001** | 0.367 | 0.838 | **<0.001** |
| SR1 | 5.25 | 0.17 | 0.51 | 0.18 | **<0.001** | 0.976 | 0.609 | 0.326 |
| Synergistetes | 0.90 | 1.44 | 5.19 | 4.57 | **<0.001** | **<0.001** | 0.838 | n. d. |
| Tenericutes | 1.02 | 0.18 | 0.21 | 0.13 | **0.011** | 0.976 | 0.060 | **0.001** |
| Verrucomicrobia | 6.87 | 10.44 | 11.36 | 5.86 | 0.478 | 0.989 | **<0.001** | **<0.001** |

TP, timepoint; n. d., not defined.

The rumen families with an abundance of at least 10% in at least one sampling timepoint were Prevotellaceae, Porphyromonadaceae, Verrucomicrobia_Subdivision5_genera_incertae_sedis, and Ruminococcaceae. In total, 25 rumen bacterial families exhibited an abundance of at least 1% at at least one sampling timepoint.

Most (13 of 25 of the most abundant rumen bacterial families) significant abundance shifts occurred after the change of housing, together with the change from the diet fed at the University Farm in Žirany to the control diet (TP1 to TP2). The addition of DGP in an amount of 1% to the control diet (TP2 to TP3) caused two significant shifts by Acidaminococcaceae and Synergistaceae. After an increase in the level of DGP in the diet from 1% to 2% (TP3 to TP4), 10 significant shifts were detected, and 9 significant shifts occurred in families with a relative abundance of over 1.0% when comparing TP2 with TP4 (Table 3).

**Table 3.** Dominant families with a relative abundance of over 1.0% at one or more of the sampling timepoints.

| Family (%) | TP1 | TP2 | TP3 | TP4 | *p*-Value TP1 to TP2 | *p*-Value TP2 to TP3 | *p*-Value TP3 to TP4 | *p*-Value TP2 to TP4 |
|---|---|---|---|---|---|---|---|---|
| Acidaminococcaceae | 3.80 | 5.21 | 2.88 | 2.49 | 0.760 | **0.005** | 0.559 | **<0.001** |
| Anaerolineaceae | 0.29 | 1.28 | 0.87 | 1.30 | 0.088 | 0.994 | 0.964 | n. d. |
| Bdellovibrionaceae | 1.56 | 0.33 | 0.97 | 0.22 | **0.009** | 0.994 | **0.016** | 0.332 |
| Bifidobacteriaceae | 2.44 | 0.00 | 0.00 | 0.02 | **<0.001** | 0.994 | 0.964 | 0.738 |
| Chitinophagaceae | 1.40 | 0.40 | 0.13 | 0.43 | **0.005** | 0.487 | 0.051 | 0.719 |
| Fibrobacteraceae | 1.64 | 0.46 | 0.64 | 0.25 | **<0.001** | 0.994 | **0.003** | 0.222 |
| Flavobacteriaceae | 1.20 | 1.00 | 0.87 | 1.20 | 0.453 | 0.994 | 0.559 | 0.791 |
| Gracilibacteraceae | 1.10 | 1.23 | 1.75 | 1.69 | 0.657 | 0.994 | 0.964 | n. d. |
| Lachnospiraceae | 3.35 | 2.46 | 2.38 | 3.89 | **0.001** | 0.994 | **<0.001** | **0.011** |
| Marinilabiliaceae | 1.65 | 2.34 | 1.97 | 1.05 | 0.760 | 0.994 | **0.001** | **<0.001** |
| Methanobacteriaceae | 1.07 | 1.06 | 1.53 | 6.85 | 0.981 | 0.994 | **<0.001** | **<0.001** |
| Pasteurellaceae | 0.43 | 0.13 | 1.00 | 0.04 | 0.099 | 0.348 | **0.010** | n. d. |
| Porphyromonadaceae | 8.39 | 23.11 | 17.75 | 22.23 | **0.006** | 0.866 | 0.509 | **0.004** |
| Prevotellaceae | 32.30 | 28.80 | 33.38 | 24.44 | **0.002** | 0.387 | **0.003** | 0.796 |
| Prolixibacteraceae | 0.45 | 1.00 | 0.55 | 1.10 | 0.387 | 0.866 | **0.009** | 0.981 |
| Rikenellaceae | 1.58 | 3.61 | 2.80 | 3.45 | **0.001** | 0.994 | 0.723 | 0.875 |
| Ruminococcaceae | 11.23 | 6.50 | 5.21 | 5.12 | **0.004** | 0.994 | 0.964 | 0.745 |
| Saccharibacteria_genera_incertae_sedis | 2.98 | 0.30 | 0.34 | 1.97 | **<0.001** | 0.994 | 0.062 | **0.002** |
| Sphingobacteriaceae | 0.52 | 0.49 | 0.80 | 1.35 | 0.970 | 0.866 | **0.032** | 0.705 |
| Spirochaetaceae | 1.59 | 0.82 | 0.08 | 0.11 | **<0.001** | 0.387 | 0.964 | **<0.001** |
| SR1_genera_incertae_sedis | 5.25 | 0.17 | 0.51 | 0.18 | **<0.001** | 0.994 | 0.964 | 0.413 |
| Synergistaceae | 0.09 | 1.44 | 5.19 | 4.57 | **<0.001** | **<0.001** | 0.964 | **<0.001** |
| Syntrophorhabdaceae | 0.00 | 0.02 | 0.63 | 2.71 | 0.981 | 0.994 | 0.509 | 0.280 |
| Veillonellaceae | 3.00 | 4.18 | 2.89 | 2.84 | 0.803 | 0.994 | 0.964 | 0.997 |
| Verrucomicrobia_Subdivision5_genera_incertae_sedis | 6.79 | 10.23 | 10.37 | 4.99 | 0.466 | 0.994 | **0.001** | **0.023** |

TP, timepoint; n. d., not defined.

The rumen genera with an abundance of at least 5% at at least one sampling timepoint were *Prevotella*, *Verrucomicrobia_Subdivision5_genera_incertae_sedis*, *Acetobacteroides*, *Succiniclasticum*, *Fretibacterium*, and *SR1_genera_incertae_sedis*. In total, 38 rumen bacterial genera revealed abundances of at least 0.9% at at least one of the sampling timepoints (Table 4). The change of housing and the change in diet between TP1 and TP2 caused a significant abundance shift in 19 of the 38 most abundant rumen genera. The addition of 1% DGP to the control diet (TP2 to TP3) resulted in increases in the abundance of *Paraprevotella*, *Chelonobacter*, *Fretibacterium*, and *Verrucomicrobia_Subdivision3_genera_incertae_sedis*, as well as decreases in the abundance of *Barnesiella*, *Tannerella*, *Succiniclasticum*, and *Selenomonas*. Increases in DGP, from 1% to 2% (TP3 to TP4), resulted in increases in the abundance of *Methanobrevibacter*, *Barnesiella*, and *Butyrivibrio*, as well as decreases in the abundance of *Mangroviflexus*, *Alloprevotella*, *Paraprevotella*, *Fibrobacter*, *Schwartzia*, *Chelonobacter*, and *Verrucomicrobia_Subdivision5_genera_incertae_sedis*. Twelve significant shifts occurred in the relative abundance of dominant genera, when comparing TP2 with TP4 (Table 4).

**Table 4.** Dominant genera with relative abundance over 0.9% at one or more of the sampling timepoints.

| Genera (%) | TP1 | TP2 | TP3 | TP4 | *p*-Value TP1 to TP2 | *p*-Value TP2 to TP3 | *p*-Value TP3 to TP4 | *p*-Value TP2 to TP4 |
|---|---|---|---|---|---|---|---|---|
| *Acetobacteroides* | 0.45 | 10.57 | 9.23 | 10.80 | **<0.001** | 0.994 | 0.650 | 0.928 |
| *Alloprevotella* | 0.92 | 0.43 | 0.61 | 0.10 | **<0.001** | 0.994 | **<0.001** | 0.139 |
| *Barnesiella* | 1.16 | 3.43 | 0.94 | 4.24 | 0.148 | **0.007** | **<0.001** | 0.560 |
| *Bifidobacterium* | 2.44 | 0.00 | 0.00 | 0.02 | **<0.001** | 0.994 | 0.954 | 0.825 |
| *Butyrivibrio* | 0.73 | 0.66 | 1.26 | 2.12 | 0.096 | 0.099 | **<0.001** | **<0.001** |
| *Centipeda* | 0.31 | 1.15 | 0.64 | 0.70 | 0.148 | 0.994 | 0.985 | **<0.001** |
| *Clostridium IV* | 2.05 | 0.94 | 0.19 | 0.34 | **<0.001** | 0.081 | 0.242 | 0.097 |
| *Chelonobacter* | 0.34 | 0.00 | 1.00 | 0.02 | **<0.001** | **<0.001** | **0.001** | 0.400 |
| *Falsiporphyromonas* | 2.93 | 4.97 | 6.17 | 4.54 | 0.913 | 0.253 | 0.588 | **0.003** |
| *Fibrobacter* | 1.64 | 0.46 | 0.64 | 0.25 | **<0.001** | 0.994 | **0.012** | 0.079 |
| *Flavonifractor* | 1.19 | 0.63 | 0.91 | 0.45 | 0.400 | 0.994 | 0.954 | n. d. |
| *Fretibacterium* | 0.04 | 1.36 | 5.15 | 4.56 | **<0.001** | **<0.001** | 0.954 | **<0.001** |
| *Gracilibacter* | 1.05 | 1.21 | 1.72 | 1.68 | 0.594 | 0.984 | NA | n. d. |
| *Mangroviflexus* | 0.26 | 1.09 | 1.03 | 0.30 | **<0.001** | 0.994 | **<0.001** | **<0.001** |
| *Methanobrevibacter* | 1.07 | 1.06 | 1.53 | 6.84 | 0.873 | 0.994 | **<0.001** | **<0.001** |
| *Methanomassiliicoccus* | 0.27 | 0.18 | 0.66 | 0.90 | 0.647 | 0.692 | 0.808 | 0.057 |
| *Mucinivorans* | 1.58 | 3.55 | 2.75 | 3.41 | 0.067 | 0.994 | 0.422 | 0.441 |
| *Ornatilinea* | 0.29 | 1.12 | 0.87 | 1.29 | 0.319 | 0.994 | 0.954 | n. d. |
| *Oscillibacter* | 0.75 | 1.50 | 0.52 | 0.46 | 0.924 | 0.994 | 0.954 | n. d. |
| *Parafilimonas* | 1.40 | 0.40 | 0.13 | 0.39 | **0.001** | 0.467 | 0.062 | n. d. |
| *Paraprevotella* | 3.33 | 2.02 | 3.38 | 1.47 | **<0.001** | **0.031** | **<0.001** | **<0.001** |
| *Petrimonas* | 0.03 | 0.68 | 0.96 | 0.77 | **<0.001** | 0.185 | 0.097 | n. d. |
| *Prevotella* | 28.05 | 26.35 | 29.39 | 22.86 | **<0.001** | 0.692 | 0.377 | 0.169 |
| *Pseudosphingobacterium* | 0.46 | 0.42 | 0.74 | 1.04 | 0.396 | 0.902 | 0.327 | n. d. |
| *Ruminococcus* | 1.85 | 0.43 | 0.46 | 0.27 | **<0.001** | 0.976 | 0.741 | 0.361 |
| *Saccharibacteria_genera_incertae_sedis* | 2.98 | 0.30 | 0.34 | 1.97 | **<0.001** | 0.994 | 0.062 | **0.008** |
| *Saccharofermentans* | 2.80 | 0.87 | 0.91 | 1.16 | **<0.001** | 0.994 | 0.479 | 0.376 |
| *Schwartzia* | 0.85 | 0.80 | 1.09 | 0.36 | **0.027** | 0.976 | **0.002** | **<0.001** |
| *Selenomonas* | 1.31 | 1.84 | 0.13 | 0.20 | 0.373 | **0.009** | 0.741 | **<0.001** |
| *Sporobacter* | 0.40 | 0.57 | 1.09 | 1.11 | 0.873 | 0.994 | 0.954 | 0.155 |
| *SR1_genera_incertae_sedis* | 5.25 | 0.17 | 0.51 | 0.18 | **<0.001** | 0.994 | 0.954 | 0.629 |
| *Succiniclasticum* | 3.78 | 5.10 | 2.88 | 2.48 | 0.755 | **0.001** | 0.954 | **<0.001** |
| *Syntrophorhabdus* | 0.00 | 0.02 | 0.63 | 2.71 | 0.924 | 0.994 | 0.422 | 0.365 |
| *Tannerella* | 3.30 | 2.63 | 0.30 | 1.71 | 0.179 | **0.016** | 0.077 | n. d. |
| *Treponema* | 1.44 | 0.80 | 0.07 | 0.10 | **0.001** | 0.383 | 0.954 | **<0.001** |
| *Vampirovibrio* | 1.56 | 0.33 | 0.97 | 0.22 | **0.005** | 0.994 | 0.059 | 0.194 |
| *Verrucomicrobia _Subdivision3_genera_incertae_sedis* | 0.00 | 0.06 | 0.98 | 0.85 | 0.265 | **<0.001** | 0.954 | n. d. |
| *Verrucomicrobia _Subdivision5_genera_incertae_sedis* | 6.79 | 10.23 | 10.37 | 4.99 | 0.924 | 0.994 | **0.003** | n. d. |

TP, timepoint; n. d., not defined.

Diversity indices were significantly affected by the timepoint (richness *p* = 0.000014; Shannon *p* = 0.00336); individual effects were not significant in any index. The addition of DGP into the wethers' diets increased richness in the rumen bacterial community. The average values of rumen bacterial community diversity indices and comparisons between timepoints are summarized in Table 5.

**Table 5.** Rumen bacterial community diversity indices according to sampling timepoint.

| Indices (Mean ± SD) | TP1 | TP2 | TP3 | TP4 |
|---|---|---|---|---|
| Richness | 275 ± 39 [a] | 144 ± 45 [b] | 192 ± 30 [c] | 204 ± 75 [c] |
| Evenness | 0.88 ± 0.023 [ab] | 0.90 ± 0.013 [a] | 0.89 ± 0.013 [ab] | 0.87 ± 0.047 [b] |
| Shannon | 7.17 ± 0.32 [a] | 6.39 ± 0.46 [b] | 6.73 ± 0.21 [ab] | 6.61 ± 0.74 [b] |

TP, timepoint; SD, standard deviation; [a,b,c], values within the same row bearing different superscript letters are significantly different at *p* < 0.05.

Beta diversity analysis performed using NMDS based on unifrac distances showed a clear separation of samples based on timepoint (Figure 3). Timepoint 1 samples were clustered as a separated group, and differences among other timepoints were lower. According to the PERMANOVA, 28% of the variability in the microbiome composition of ruminal fluid was explained by the timepoint ($p = 0.001$), whereas differences between individual wethers explained 22% of the variability ($p = 0.002$).

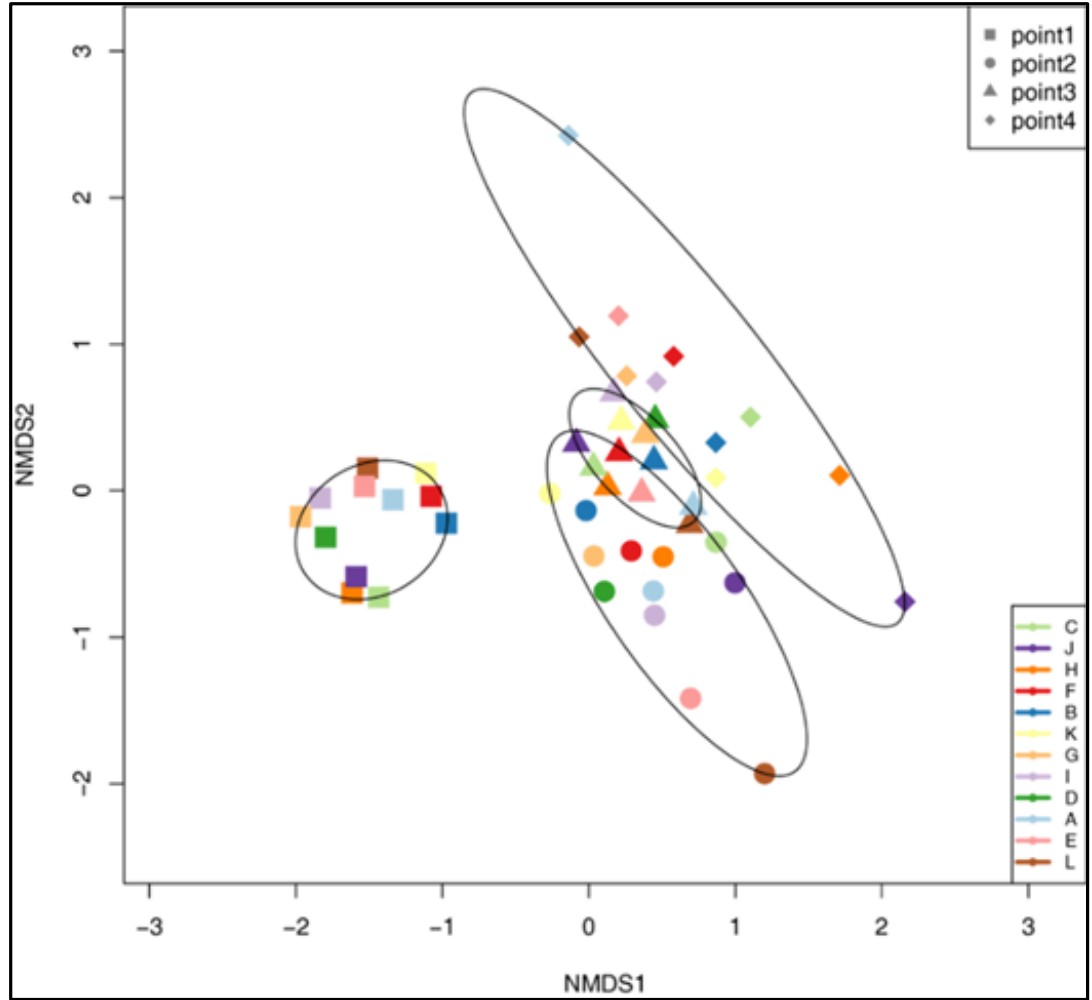

**Figure 3.** Non-metric multidimensional scaling (NMDS) plot of microbial community composition in the rumen fluid samples, according to sampling timepoints.

In silico metabolic predictions using Picrust2 were performed to determine metagenome functions. Figure 4 shows the predicted functions, which were a result of changes in the diet composition. The addition of DGP into the wethers' diets increased the amino acid degradation pathway, as well as acetone, menaquinol, and methane production, but decreased pathways for the degradation of fucose. Pairwise ANOSIM comparison showed that all timepoints significantly affected the composition; all $p$ values were 0.001. The strongest difference estimated by the R value was between timepoints 1 and 3, whereas differences between timepoints 2 and 3 and timepoints 3 and 4 were lower (Table 6).

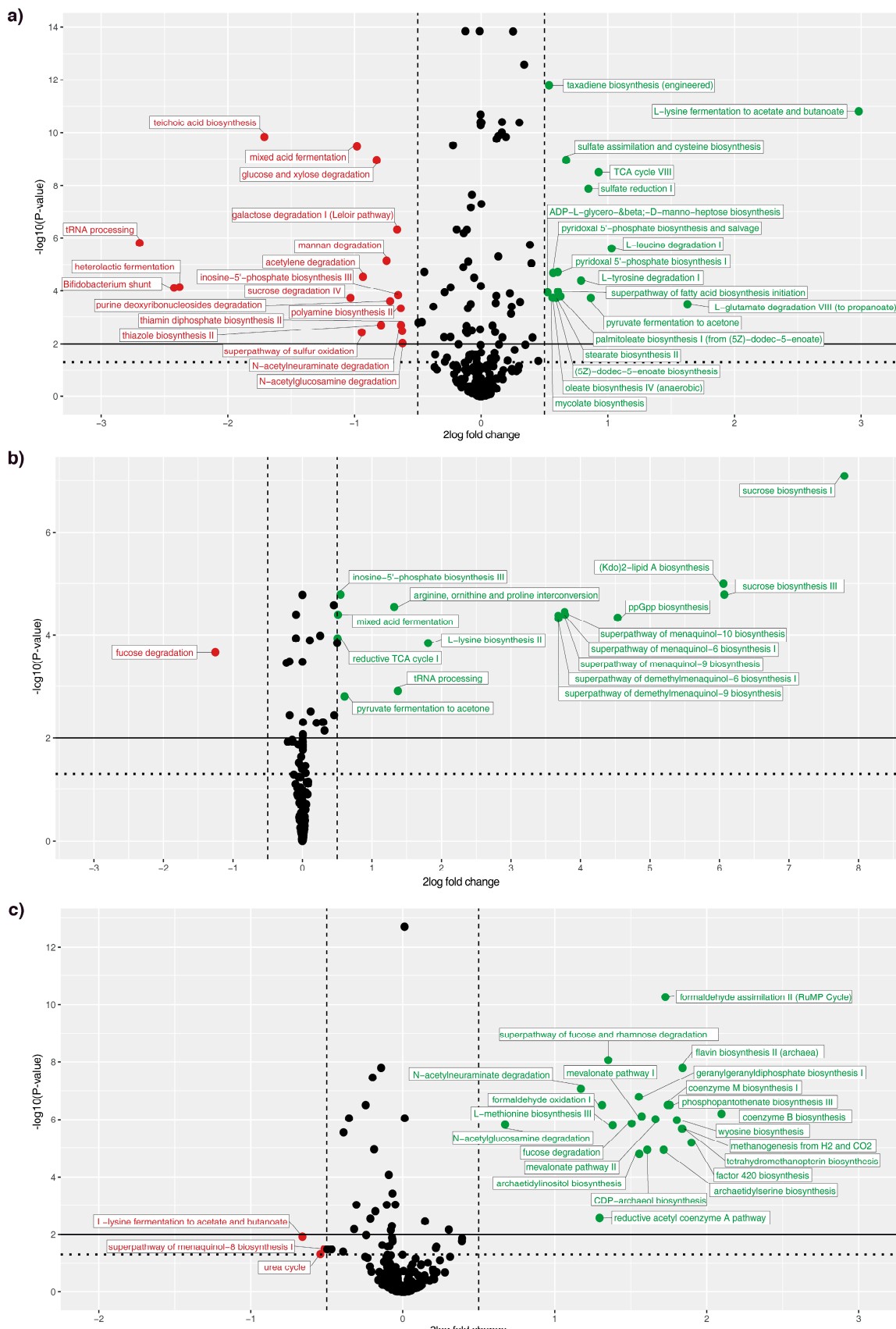

**Figure 4.** Volcano plot of Deseq analysis of metabolic pathways predicted by Picrust2 and annotated by MetaCYC, between: (**a**) TP1 and TP2; (**b**) TP2 and TP3; and (**c**) TP3 and TP4.

**Table 6.** ANOSIM comparison of rumen microbial community sampling timepoints.

| ANOSIM | TP1 | TP2 | TP3 | TP4 |
|---|---|---|---|---|
| TP1 | | *p* = 0.001 | *p* = 0.001 | *p* = 0.001 |
| TP2 | R = 0.819 | | *p* = 0.001 | *p* = 0.001 |
| TP3 | R = 0.958 | R = 0.306 | | *p* = 0.001 |
| TP4 | R = 0.934 | R = 0.588 | R = 0.320 | |

TP, timepoint.

## 4. Discussion

Bacteroidetes and Firmicutes have been reported to be the dominant rumen bacterial phyla [25]. Li et al. [26] and Liu et al. [27] reported that Bacteroidetes and Firmicutes were the most abundant sheep rumen phyla in addition to Proteobacteria, Verrucomicrobiota, Spirochaetes, and Fibrobacteres, which is similar to the results of this study. Additionally, in the study by Liu et al. [27], *Prevotella* was detected as the predominant sheep rumen genus. The addition of DGP into the diet did not affect the abundance of *Prevotella* (Table 4). According to Liu et al. [27], *Prevotella* has high genetic variability, which enables this genus to occupy different ecological niches within the rumen. It seems that the abundance of *Prevotella* remains unchanged after the intake of polyphenol-rich feed, such as a daily diet containing DGP in amounts of 1% or 2%. *Succiniclasticum* has mainly been identified in the liquid fraction of the rumen, and is specialized in fermenting succinate to propionate [12,28]. Diaz Carrasco et al. [28] demonstrated an increase (from 1.99% to 3.99%) in *Succiniclasticum* in the rumen after the supplementation of a diet with tannin at a concentration of 2 g per kilogram of feed. This contrasted with the results of this study, where the abundance of *Succiniclasticum* decreased from 5.10% to 2.88% (*p* = 0.001) after the ingestion of feed containing a polyphenol, i.e., DGP. Similar differences in results were also seen with *Selenomonas*, which are starch- and sugar-degrading bacteria and possess lipolytic activity. In this study, higher abundances of *Butyrivibrio* and *Methanobrevibacter* were determined in rumen fluid after the addition of grape pomace into the wethers' diets. *Butyrivibrio*, a bacterium involved in fiber, pectin, and urea degradation in the rumen, was shown to increase in this study after the addition of DGP to the diet. It can be hypothesized that the addition of DGP into the diet provides substances supporting the growth of these two genera in the rumen. However, Diaz Carrasco et al. [28] claimed that the abundance of *Butyrivibrio* was negatively affected by tannin treatment. It is known that grape pomace contains tannins, as well as other bioactive substances [29] which, in adequate concentrations, can have a positive influence. However, the increased intake of DGP (from 1% to 2%) had a significantly negative effect on other rumen cellulolytic bacteria. The abundance of *Ruminococcus* decreased from 0.46% to 0.27%, and the abundance of *Fibrobacter* decreased from 0.64% to 0.25% (Table 4). Decreases in these rumen cellulolytic bacteria after the increased intake of feed containing polyphenols, such as tannins, are in accordance with the findings of Vasta et al. [12], Frutos et al. [30], and Jones et al. [31]. Ma et al. [32] concluded that the addition of 25 mg of resveratrol to the diet significantly reduced the abundance of *Methanobrevibacter*, which resulted in decreased methane production [32,33]. Tavendale et al. [34] found that tannins completely inhibited methane production by limiting the growth of *Methanobrevibacter* strains YLM-1 and DSM1093. Additionally, Newbold and Ramos-Morales [35] claimed that plant polyphenol compounds have demonstrated potential to decrease methane emissions from ruminants. In this study, the intake of feed with increased amounts of DGP, which contains several polyphenols, including tannin and resveratrol, resulted in upregulation of the methane production pathway. This finding was supported by an increase in methanogens, such as *Methanobrevibacter*, but also by a decrease in the genus *Ruminococcus* (Table 4), which, according to Ross et al. [36], is important for decreasing methane after the addition of grape pomace to the diet.

The greatest number of significant changes in the abundance of rumen phyla, families, and genera were observed following changes in housing and diet (TP1 to TP2) (Tables 2–5).

A change in housing conditions and diet composition will cause a change in the rumen bacterial community. Such findings have also been published by Wu et al. [37] and Henderson et al. [38].

The change of the housing and the diet fed at the University Farm in Žirany to the housing and the control diet fed during the experiment (TP1 to TP2) decreased the richness, as well as the Shannon index, of the rumen bacterial community (Table 5). The addition of DGP changed the richness and evenness diversity indices, whereas the Shannon index remained unchanged as compared with the control diet (Table 5). Liu et al. [39] experimented with a diet with a similar concentrate-to-forage ratio as that used in this study. They determined the Shannon index to be >8, which is a much higher result compared with the control diet and the diet containing DGP assessed in this study. Li et al. [26] found the Shannon index to be 5.48 for lambs with a low subacute ruminal acidosis risk. Belanche et al. [40] determined a Shannon index of 6.35 and evenness of 0.83 for grazing sheep. These results are comparable with the results of this study. However, Langda et al. [41] estimated a Shannon index of 5.5 for grazing sheep, which was associated with the richness and evenness of rumen bacteria; this was lower compared with the results of this study. A stable rumen bacterial community will provide beneficial effects on the health and performance of ruminants. The addition of DGP into the wethers' diet increased the richness of the rumen bacterial community, which is favorable for maintaining rumen bacterial homeostasis. Vašeková et al. [42] determined total polyphenols to be 27.38 mg GAE/g in the DGP diets which were fed to wethers in this study. Consequently, the calculated intakes of total polyphenols in the diets containing 1% or 2% DGP were 282 or 564 mg GAE, respectively, which were higher than the control diet.

The results of this study, as well as comparison with other published articles, indicate the various effects of specific polyphenols on rumen bacteria. However, a previous study [43] supported the hypothesis that including grape pomace in the diet of ruminants would increase organic matter and neutral detergent fiber digestibility. From this point of view, the addition of DGP has a positive effect on the performance of ruminants; however, the costs of drying fresh grape pomace may be problematic.

## 5. Conclusions

A change in housing, as well as addition of dried grape pomace, significantly affected the rumen bacterial phyla, families, and genera, as well as the rumen bacterial community indices. No adverse effect of the addition of dried grape pomace on the rumen bacterial community was noted. Future studies should investigate exactly which species and strains are susceptible to intakes of specific polyphenols and what concentrations or combinations of polyphenols are the most beneficial for rumen bacteria.

**Author Contributions:** Conceptualization, M.R., J.M. and M.J.; data curation, M.R., J.M., M.G. and M.M.; formal analysis, J.M., M.G., M.M. and Z.S.; funding acquisition, B.G.; investigation, M.R.; methodology, M.G., D.B. and M.J.; project administration, B.G. and Z.S.; resources, M.R. and B.G.; software, O.H., Z.S. and L.Z.; supervision, D.B.; validation, M.Š., B.G. and D.B.; visualization, J.M. and Z.S.; writing—original draft preparation, M.R. and J.M.; writing—review and editing, O.H., L.Z. and M.J. All authors have read and agreed to the published version of the manuscript.

**Funding:** This study and the APC were funded by the Slovak Research and Development Agency, grant number APVV-16-0170 (by-products from grape processing as a bioactive substance source in animal nutrition).

**Institutional Review Board Statement:** The conditions of animal care, manipulations, and use adhered to guidance of the Ethics Committee of the Slovak University of Agriculture in Nitra, Protocol No. 48/2013. According to the State Veterinary and Food Administration of the Slovak Republic, the given study has been evaluated as an inexperienced agricultural practice that does not fall under the legislation of Government Regulation of the Slovak Republic 377/2012 of 14 November 2012, laying down requirements for the protection of animals used for scientific or educational purposes.

**Data Availability Statement:** Not applicable.

**Acknowledgments:** The authors would like to acknowledge the University Farm (Slovak University of Agriculture in Nitra, Slovakia) for establishing the appropriate conditions for this experiment.

**Conflicts of Interest:** The authors declare no conflict of interest.

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
