# Peer review of "Effect of Grape Pomace Intake on the Rumen Bacterial Community of Sheep"

_diversity, doi:10.3390/d15020234_

Round 1

Reviewer 1 Report

This is another interesting study which investigates the addition of a substance (in this case grape pomace) to the diet of animals and the effect which it has on the gut microbiota. The reasoning behind the study is sound, and it appears to be executed well, although a bit more data in the methodology would be beneficial (detailed below).

I am imagining that the authors first language is no English, and this shows within the manuscript. I strongly recommend sending future manuscripts to a native English speaker.

I have tried to detail some of the required changes below to make the manuscript read appropriately. These are written as the suggested change.

Line 18- properties of the rumen bacterial community ….

Line 19- fed to the ruminants. ….

Line 21- Rumen fluid samples from 12 wethers were used for 16S rRNA gene sequencing and subsequent bacterial identification.

Line 22- At the phylum level, Bacteriodetes and Firmicutes, from the family level Prevotella and Porphyromonadaeceae and at the genera level, Prevotella and Verrucomicrobia … were the most common bacteria, regardless of the diet which the animals were fed on.

Line 30- Results of this study shows a significant effect ….

Line 32- In addition, a diet supplemented with …

Line 32- revelated higher activity of the methoanogenesis pathway in the rumen.

Line 38- maintenance of a stable rumen ….

Line 40- protein in the rumen. …..

Line 42- alteration of the rumen bacterial ……

Line 42- If this is a negative alteration, the efficiency  …..

Line 44- factor affecting the rumen bacterial ….

Line 47-48- affects the amount of ….

Line 52- found an increase in the diversity of the rumen microbiota ….

Line 53- considering that research on the dietary effects on the rumen microbiota is highly topical, and the results of experiments with grape pomace and the gut microbiota are not uniform, this study was designed to address some of these issues.

Line 56- The effect of dietary change, and the addition of grape pomace on the bacterial community of the rumen was investigated.

Line 60- more detail needed here- what breed were they? Age? Etc

Line 61- wethers were marked as an individual …..

Line62- before the experiment in the University Farm ……

Line 63- The experiment took place at the barn ….

Line 65- In each period of the experiment …..

Line 66- of the animals genetic make up, condition and  ……

Line 67- rumen fluid samples were collected ……

Line 69- where the experiment was carried out …..Line 69- after the first sample collection ….

Line 70- with a control diet …..

Line 71- After the end of the preparatory period, a second rumen fluid…..

Line 72- were fed for 12 days with a control diet ….

Line 73- please define DGP.

Line 73- During the last day of the experimental …..

Line 74- Then the 12 day long experimental period started and animals were fed with a control diet containing 2% DGP.

Line 75- During the last day of the second experiment ….

Line 77- All diets, mineral and vitamin lick and water were available ad libitum throughout the entire study period.

Line 80- To ensure that only feed intake from the tested diets occurred, from the start of the preparatory period to the end of the second experimental …..

Line 80-81- will this likely stress the animals and lead to altered results?

Why is figure 1 a scheme and not labelled figure 1?

Line 101- collected on the morning of the last day ….

Line 101- Collection of 2ml of rumen fluid occurred after fixation of the wether using ….

Line 111- please include manufacturer

Line 114- checked on an agarose gel ….

Line 116- libraries were prepared ….

Line 124- metabolic pathways annotated by …..

Line 131- using a 2-factor model ….

Line 140- ASVs were common for all …..

Why is figure 2 a scheme and not labelled figure 2?

Line 148- were the most abundant in rumen fluid …..

Line 152- The phyla with an abundance of at least 1% in at least one sampling timepoint were ……

Line156- The change from the diet fed at the University Farm in Zirany to the control diet (TP1 to TP2) caused a shift in relative abundance of some bacteria which was significant for 9 of 12 of the most abundant phyla.

Table 2- you have to make it clear in the legend that these only had to be over 1% in one or more of the sampling times.

Line 170- revealed an abundance of at least 1% …..

Line 171- Most (13 of 25 of the most ….)

Line 172- after the change from the diet fed at the University farm in Zirany to the control diet ….

Line 174- increase of DGP in the diet from ….

Line 189- caused a significant abundance shift in 19 of 38 of the most abundant rumen genera…..

Line 215-  clustered as a separate group ….

Why is figure 3 a scheme and not labelled figure 3?

Also Figure 3 is too small to see, these need to be bigger to be interpreted

Line 232- In silico needs to be in italics

Line 235- showed instead of shoved

Line 236- The strongest difference estimated by the R value was …..

Line 240- are listed as the dominant rumen bacterial phyla …..

Line 241- Li et al [21] and Liu et al [22] reported that Bacteriodetes and Firmicutes were the most abundant sheep rumen phyla along with ……

Line 243- Also in a study by …

Line 244- detected as the predominant sheep …..

Line 244- did not affect the abundance ….

Line 248- rich feeds such as DGP in an ….

Line 249- mainly in the liquid fraction of the rumen ….

Line 250- published an increase …..

Line 251- in the rumen after ….

Line 251- with a tannin at a concentration of 2g per kg …

Line 252- this is a difference with the results of this study …..

Line 254- similar differences in results were also seen with Selenomonas …..

Line 258- degradation in the rumen were shown to increase in this study after addition ….

Line 261- by tannin treatment …..

Line 262- but also contains may other bioactive…..

Line 263- can have a positive influence ….

Line 264- had a significant negative effect …..

Line 275- resulted in up regulation of the methane production pathway

Line 276- This finding is supported by an increase in ….

Line 277- also by a decrease in the genera ….

Line 278- Line 278- what is grape marc?

Line 279- The change from the diet fed at the University farm in Zirany to the control diet (TP1 to TP2) decreased the richness ….

Line 280- Shannon index of the rumen ….

Line 282- compared to the control diet ….

Line 282- Liu et al [32] fed a diet with ….

Line 283- as was used in this study …..

Line 284- to the control diet as well as the diet containing ….

Line 285- can be due to a different diet ….

Line 286- compounds decreased the bacterial diversity of the rumen

Line 287- This is contradictory to the statement of Ramos-Morales et al [33] in this ….

Line 288- into the control diet ….

Line 289- decreased in the diet containing ….

Line 290- a Shannon index of 5.48 for lambs with …

Line 291- Belanche et al., [34] determined a Shannon index of 6.35 and evenness of …..

Line 293- grazing sheep a Shannon index…

Line 294- rumen bacteria of 5.5, which is lower compared to the result …..

Line polyphenols to be 27.38 mg ….

Line 296- in the diet containing ….

Line 297- respectively, which is more than the control diet. …..

Line 297- The results of this study and its comparison to other published articles point out the various effects of specific polyphenols on rumen bacteria ……

Line 299 on animals range from beneficial …..

Line 300- 301- delete this sentence as it is a repeat of previous words

Line 302- articles investigating the rumen bacterial community point out the inconsistent and adverse effects of using polyphenol …..

Author Response

Many thanks for reviewing and commenting on our article as well as correcting the flaws in our English grammar. All comments have been accepted and incorporated into the text, which has improved the quality of this article.

Line 18- properties of the rumen bacterial community ….

Response: corrected in the text

Line 19- fed to the ruminants. ….

Response: corrected in the text

Line 21- Rumen fluid samples from 12 wethers were used for 16S rRNA gene sequencing and subsequent bacterial identification.

Response: corrected in the text

Line 22- At the phylum level, Bacteriodetes and Firmicutes, from the family level Prevotella and Porphyromonadaeceae and at the genera level, Prevotella and Verrucomicrobia … were the most common bacteria, regardless of the diet which the animals were fed on.

Response: corrected in the text

Line 30- Results of this study shows a significant effect ….

Response: corrected in the text

Line 32- In addition, a diet supplemented with …

Response: corrected in the text

Line 32- revelated higher activity of the methoanogenesis pathway in the rumen.

Response: corrected in the text

Line 38- maintenance of a stable rumen ….

Response: corrected in the text

Line 40- protein in the rumen. …..

Response: corrected in the text

Line 42- alteration of the rumen bacterial ……

Response: corrected in the text

Line 42- If this is a negative alteration, the efficiency  …..

Response: corrected in the text

Line 44- factor affecting the rumen bacterial ….

Response: corrected in the text

Line 47-48- affects the amount of ….

Response: corrected in the text

Line 52- found an increase in the diversity of the rumen microbiota ….

 Response: corrected in the text

Line 53- considering that research on the dietary effects on the rumen microbiota is highly topical, and the results of experiments with grape pomace and the gut microbiota are not uniform, this study was designed to address some of these issues.

Response: corrected in the text

Line 56- The effect of dietary change, and the addition of grape pomace on the bacterial community of the rumen was investigated.

Response: corrected in the text

Line 60- more detail needed here- what breed were they? Age? Etc

Response: Thank that you pay attention to the details. In this experiment wethers of Ile de France breed were used. All 12 wethers were selected from one herd. We have information about live weight of wethers. The average live weight at the start of experiment was 35.28±2.04 kg. These information are listed in Materials and Methods in chapter Animals and experiment design. Unfortunately, we do not have information about exact age of wethers.

Line 61- wethers were marked as an individual …..

Response: corrected in the text

Line62- before the experiment in the University Farm ……

Response: corrected in the text

Line 63- The experiment took place at the barn ….

Response: corrected in the text

Line 65- In each period of the experiment …..

Response: corrected in the text

Line 66- of the animals genetic make up, condition and  ……

Response: corrected in the text

Line 67- rumen fluid samples were collected ……

Response: corrected in the text

Line 69- where the experiment was carried out …..Line 69- after the first sample collection ….

Response: corrected in the text

Line 70- with a control diet …..

Response: corrected in the text

Line 71- After the end of the preparatory period, a second rumen fluid…..

Response: corrected in the text

Line 72- were fed for 12 days with a control diet ….

Response: corrected in the text

Line 73- please define DGP.

Response: Abbreviation of DGP is explained in the abstract /line 21/. To the information about dried grape pomace the origin was added /line 80/.

Line 73- During the last day of the experimental …..

Response: corrected in the text

Line 74- Then the 12 day long experimental period started and animals were fed with a control diet containing 2% DGP.

Response: corrected in the text

Line 75- During the last day of the second experiment ….

Response: corrected in the text

Line 77- All diets, mineral and vitamin lick and water were available ad libitum throughout the entire study period.

Response: corrected in the text

Line 80- To ensure that only feed intake from the tested diets occurred, from the start of the preparatory period to the end of the second experimental …..

Response: corrected in the text

Line 80-81- will this likely stress the animals and lead to altered results?

Response: Thank for your question. Housing these animals in a pen with bedding (for example straw) will deteriorate whole experiment. Wethers would intake this straw ad libitum – in that case, the straw will be a part of the daily diet in the unknown amount. And the rumen bacterial community will be changed. To prevent the effect of unknown straw (or other bedding) intake, animals were during this study housed without bedding. Housing without bedding will not altered the results.

Why is figure 1 a scheme and not labelled figure 1?

Response: corrected in the text

Line 101- collected on the morning of the last day ….

Response: corrected in the text

Line 101- Collection of 2ml of rumen fluid occurred after fixation of the wether using ….

Response: corrected in the text

Line 111- please include manufacturer

Response: Manufacturer was added into the text.

Line 114- checked on an agarose gel ….

Response: corrected in the text

Line 116- libraries were prepared ….

Response: corrected in the text

Line 124- metabolic pathways annotated by …..

Response: corrected in the text

Line 131- using a 2-factor model ….

Response: corrected in the text

Line 140- ASVs were common for all …..

Response: corrected in the text

Why is figure 2 a scheme and not labelled figure 2?

Response: corrected in the text

Line 148- were the most abundant in rumen fluid …..

Response: corrected in the text

Line 152- The phyla with an abundance of at least 1% in at least one sampling timepoint were ……

Response: corrected in the text

Line156- The change from the diet fed at the University Farm in Zirany to the control diet (TP1 to TP2) caused a shift in relative abundance of some bacteria which was significant for 9 of 12 of the most abundant phyla.

Response: corrected in the text

Table 2- you have to make it clear in the legend that these only had to be over 1% in one or more of the sampling times.

Response: corrected in the text

Line 170- revealed an abundance of at least 1% …..

Response: corrected in the text

Line 171- Most (13 of 25 of the most ….)

Response: corrected in the text

Line 172- after the change from the diet fed at the University farm in Zirany to the control diet ….

Response: corrected in the text

Line 174- increase of DGP in the diet from ….

Response: corrected in the text

Line 189- caused a significant abundance shift in 19 of 38 of the most abundant rumen genera…..

Response: corrected in the text

Line 215-  clustered as a separate group ….

Response: corrected in the text

 Why is figure 3 a scheme and not labelled figure 3?

Response: corrected in the text

Also Figure 3 is too small to see, these need to be bigger to be interpreted

Response: Answer: Thank you for your note. After revision Figure 3. is now Figure 4. For better interpretation Figure 4. will be attached to the article in (PDF) format.

Line 232- In silico needs to be in italics

Response: corrected in the text

Line 235- showed instead of shoved

Response: corrected in the text

Line 236- The strongest difference estimated by the R value was …..

Response: corrected in the text

Line 240- are listed as the dominant rumen bacterial phyla …..

Response: corrected in the text

Line 241- Li et al [21] and Liu et al [22] reported that Bacteriodetes and Firmicutes were the most abundant sheep rumen phyla along with ……

Response: corrected in the text

Line 243- Also in a study by …

Response: corrected in the text

Line 244- detected as the predominant sheep …..

Response: corrected in the text

Line 244- did not affect the abundance ….

Response: corrected in the text

Line 248- rich feeds such as DGP in an ….

Response: corrected in the text

Line 249- mainly in the liquid fraction of the rumen ….

Response: corrected in the text

Line 250- published an increase …..

Response: corrected in the text

Line 251- in the rumen after ….

Response: corrected in the text

Line 251- with a tannin at a concentration of 2g per kg …

Response: corrected in the text

Line 252- this is a difference with the results of this study …..

Response: corrected in the text

Line 254- similar differences in results were also seen with Selenomonas …..

Response: corrected in the text

Line 258- degradation in the rumen were shown to increase in this study after addition ….

Response: corrected in the text

Line 261- by tannin treatment …..

Response: corrected in the text

Line 262- but also contains may other bioactive…..

Response: corrected in the text

Line 263- can have a positive influence ….

Response: corrected in the text

Line 264- had a significant negative effect …..

Response: corrected in the text

Line 275- resulted in up regulation of the methane production pathway

Response: corrected in the text

Line 276- This finding is supported by an increase in ….

Response: corrected in the text

Line 277- also by a decrease in the genera ….

Response: corrected in the text

Line 278- Line 278- what is grape marc?

Response:  Thank you for your comment. After definition grape pomace and grape marc are the same by-product of grape processing. However, some authors defined grape marc as a grape pomace without grape seeds.

Line 279- The change from the diet fed at the University farm in Zirany to the control diet (TP1 to TP2) decreased the richness ….

Response: corrected in the text

Line 280- Shannon index of the rumen ….

Response: corrected in the text

Line 282- compared to the control diet ….

Response: corrected in the text

Line 282- Liu et al [32] fed a diet with ….

Response: corrected in the text

Line 283- as was used in this study …..

Response: corrected in the text

Line 284- to the control diet as well as the diet containing ….

Response: corrected in the text

Line 285- can be due to a different diet ….

Response: corrected in the text

Line 286- compounds decreased the bacterial diversity of the rumen

Response: corrected in the text

Line 287- This is contradictory to the statement of Ramos-Morales et al [33] in this ….

Response: corrected in the text

Line 288- into the control diet ….

Response: corrected in the text

Line 289- decreased in the diet containing ….

Response: corrected in the text

Line 290- a Shannon index of 5.48 for lambs with …

Response: corrected in the text

Line 291- Belanche et al., [34] determined a Shannon index of 6.35 and evenness of …..

Response: corrected in the text

Line 293- grazing sheep a Shannon index…

Response: corrected in the text

Line 294- rumen bacteria of 5.5, which is lower compared to the result …..

Response: corrected in the text

Line polyphenols to be 27.38 mg ….

Response: corrected in the text

Line 296- in the diet containing ….

Response: corrected in the text

Line 297- respectively, which is more than the control diet. …..

Response: corrected in the text

Line 297- The results of this study and its comparison to other published articles point out the various effects of specific polyphenols on rumen bacteria ……

Response: corrected in the text

Line 299 on animals range from beneficial …..

Response: corrected in the text

Line 300- 301- delete this sentence as it is a repeat of previous words

Response: corrected in the text

Line 302- articles investigating the rumen bacterial community point out the inconsistent and adverse effects of using polyphenol …..

Response: corrected in the text

Reviewer 2 Report

This study focused on the effects of dried grape pomace (DGP) on rumen bacterial community of sheep. 16S rRNA gene sequencing and bacterial identification were performed for different concentrations of DGP addition.The results show that diet supplemented with DGP has an effect on the abundance of some rumen bacteria and diversity indices. The design of this study is relatively simple and reasonable. However, the practical application value was not clearly written. The sections about results and discussion are lack of focus. Conclusions should be supplemented. Some details need to be noticed, and the description should be more concise and logical. The structure of this article is not very well-organized and needs significant adjustment.

Abstract:

1. The experimental results in the abstract should be described in more important parts of the research. (Line 25-28) .

2. Lack of practical application significance of this study. (Line 31-33)

3. Only genera and species should be written in italics, and others in regular. (Line 22-24)

4. The result conclusion (line 30-33) seems repetitive with the previous statement (line 22-28).

Introduction:

5. Lack of detailed background on wine industry by-products. The current application of grape pomace in feed addition can be added in this part. (Line 45-46)

6. The word “increase“ is supposed to be “increasing“. There are many other spelling and grammatical mistakes in the text. Please check them carefully. (Line 52-53)

Materials and methods:

7. What are the reasons for choosing different sites for feeding? The significance of this experiment should be stated separately. The determination of timepoint 1 does not seem to be relevant for this experiment. (Line 67-69)

8. The word firs is supposed to be first. (Line 70)

9. The basis and experimental significance for the selection of different concentrations of DGP should be added. (Line 72-75)

Results:

10. “the high portion of ASVs is specific for timepoint 1 and timepoint 2.” is inconsistent with the Scheme 2. The significance of testing at both time points should be stated. (Line 138-139)

11. The data already provided in the table does not need to be provided here. (Line 146-161)

12. The Phylum, family and genera can be arranged alphabetically for ease of reading. (Table 2, 3, 4)

13. The comparison between TP2 and TP4 should be added.

14. Lack of individual annotations in Scheme 3. (Scheme 3)

15. The word “shoved” is supposed to be “showed”. (Line 235)

Discussion:

16. The discussion section should be rewritten.The discussion should be written in sections and add content that fits the main thrust of the journal.The effects of feeding DGP on ruminants should be discussed.

17. Confusion of Causation. (Line 285-286)

18. Future perspectives about economic values of adding grape pomace to ruminant diets should be discussed.

Conclusions:

19. Conclusions should be segmented separately

Author Response

Many thanks for reviewing and commenting on our article. All comments have been accepted and incorporated into the text, which has improved the quality of this article.

Abstract:

  1. The experimental results in the abstract should be described in more important parts of the research. (Line 25-28)

Response: abstract was modified

  1. Lack of practical application significance of this study. (Line 31-33)

Response: abstract was modified

  1. Only genera and species should be written in italics, and others in regular. (Line 22-24)

Response: corrected in the text and tables

  1. The result conclusion (line 30-33) seems repetitive with the previous statement (line 22-28).

Response: repetitive statement was deleted, abstract was modified

Introduction:

  1. Lack of detailed background on wine industry by-products. The current application of grape pomace in feed addition can be added in this part. (Line 45-46)

Response: Thank for your recommendation. The information about grape pomace and its using in feeding of animals/ruminants were added to the introduction.

  1. The word “increase“ is supposed to be “increasing“. There are many other spelling and grammatical mistakes in the text. Please check them carefully. (Line 52-53)

Response: after revision, whole article was sent to MDPI English language editing services.

Materials and methods:

  1. What are the reasons for choosing different sites for feeding? The significance of this experiment should be stated separately. The determination of timepoint 1 does not seem to be relevant for this experiment. (Line 67-69)

Response: Thank you for your valuable comment. You are right, it seem, that result from timepoint 1 are not relevant for the evaluation of the effect of grape pomace feeding on the rumen bacterial community. On the other hand, the results of timepoint 1 sampling are gained from the same wethers as it was by other sampling timepoints. And one of the aims of this study was to point out the changes occurred in rumen bacterial community after change of feeding together with moving from one to another housing. We think, that also these results (comparison TP1 to TP2) are very interesting and useful for scientific community as well as for other readers.

  1. The word “firs“ is supposed to be “first“. (Line 70)

Response:  corrected in the text

  1. The basis and experimental significance for the selection of different concentrations of DGP should be added. (Line 72-75)

Response: Thank you for comment. The concentration of DGP in the daily diet of the wethers during both of experimental period was set in accordance to the methodology of research project (APVV‐16‐0170 By‐products from grape processing as a bioactive substance source in animal nutrition) within which this experiment was realised.

Results:

  1. “the high portion of ASVs is specific for timepoint 1 and timepoint 2.” is inconsistent with the Scheme 2. The significance of testing at both time points should be stated. (Line 138-139)

Response: Thank you for your note. You are indeed correct, we made a mistake in the text and there should have been a TP1 and a TP4. The Venn diagram is just to compare the sharing of each ASV.   The amount of ASVs is statistically evaluated in the diversity index table as "richness".

  1. The data already provided in the table does not need to be provided here. (Line 146-161)

Response: The values of relative abundance were deleted from the text.

  1. The Phylum, family and genera can be arranged alphabetically for ease of reading. (Table 2, 3, 4)

Response:  corrected in the text

  1. The comparison between TP2 and TP4 should be added.

Response:  Comparison between TP2 and TP4 was added in the Table 2, 3 and 4, and in the text.

  1. Lack of individual annotations in Scheme 3. (Scheme 3)

Response: After comment of reviewer 1 Scheme 3. is now Figure 4. Additional information were incorporated in the text.

  1. The word “shoved” is supposed to be “showed”. (Line 235)

Response:  corrected in the text

Discussion:

  1. The discussion section should be rewritten.The discussion should be written in sections and add content that fits the main thrust of the journal.The effects of feeding DGP on ruminants should be discussed.

Response: Thank you for your recommendation. We thing that aim of this article mainly the information about diversity indices of rumen bacterial community meets the aim and scope of journal Diversity. Information about effect of DGP on nutrients digestibility of ruminants was added into chapter Discussion. We know that rumen bacterial community affect the rumen function as well as whole animal and its effectivity of production. But this study was aimed only to detection and description of rumen bacterial community.

  1. Confusion of Causation. (Line 285-286)

Response: Thank you for your attention to details. You are indeed correct, confusing sentences were deleted.

  1. Future perspectives about economic values of adding grape pomace to ruminant diets should be discussed.

 Response:  Discussion about economic values of adding grape pomace to ruminant diets was added to the text.

Conclusions:

  1. Conclusions should be segmented separately

Response:  corrected in the text

Round 2

Reviewer 1 Report

I wish to thank the authors for diligently addressing my comments to the manuscript. 

The manuscript reads well 

Author Response

Thanks again for reviewing the article and suggesting corrections to the errors.

Reviewer 2 Report

This study focused on the effects of dried grape pomace (DGP) on rumen bacterial community of sheep. 16S rRNA gene sequencing and bacterial identification were performed for different concentrations of DGP addition. The results show that diet supplemented with DGP has an effect on the abundance of some rumen bacteria and diversity indices. The design of this study is relatively simple and reasonable. After modification, some sections are still unsatisfactory. Paragraphs are too long that readers can't quickly extract useful information from this section. It is recommended to discuss the results in groups, or use a deduction and summary format.

Keyword

1. More relevant detailed keywords like “sheep”, or “grape pomace” can be added.

Abstract

2. Genera should be written in italics (line 24-25).

3. The conclusion in abstract (line 32-35) is inconsistent with Conclusions part.

Introduction

4. Appropriate paragraphing is suggested.

5. “The effects of dietary change, and the addition of grape pomace to sheep diet on the bacterial community of the rumen was were investigated. ”(line 67-70The study purpose did not mention the effect of location, but there was a shift of location between time point 1 and 2.

Materials and Methods

6. If “Pointing out the changes occurred in rumen bacterial community after change of feeding together with moving from one to another housing.” is one of the aims of this study. Other parts should show this aim clearly, and the effect of changing housing environment should be illustrated as an influence factor in discussion instead of attributing all effect to the diet change.

7. The determination of timepoint 1 does not seem to be relevant for this experiment.

8. The word “Scheme” should be “Figure”. (Line 79)

Discussion

9. Appropriate paragraphing is suggested.

10. The effect of different concentration of DGP should be discussed.

Conclusions

11. The conclusion is confusing and inconsistent with the majority of the discussion. If the aim of the study is to figure out the effect of grape pomace, this part should include a summary about that.

Author Response

Keyword

  1. More relevant detailed keywords like “sheep”, or “grape pomace” can be added.

Response: grape pomace and sheep were added to keywords.

Abstract

  1. Genera should be written in italics (line 24-25).

 Response: genera in the abstract is now written in italics.

  1. The conclusion in abstract (line 32-35) is inconsistent with Conclusions part.

 Response: conclusions were modified. Two sentences about results of this study were added to the conclusions. Now abstract is consistent with conclusions.

Introduction

  1. Appropriate paragraphing is suggested.

Response: paragraphing was applicate in the introduction.

  1. “The effects of dietary change, and the addition of grape pomace to sheep diet on the bacterial community of the rumen was were investigated. ”(line 67-70)The study purpose did not mention the effect of location, but there was a shift of location between time point 1 and 2.

Response: the effect of dietary change together with shift of location on the bacterial community of the rumen was added into the last sentence of introduction.

Materials and Methods

  1. If “Pointing out the changes occurred in rumen bacterial community after change of feeding together with moving from one to another housing.” is one of the aims of this study. Other parts should show this aim clearly, and the effect of changing housing environment should be illustrated as an influence factor in discussion instead of attributing all effect to the diet change.

Response: Thank you for your comment on the basis of which we have added information to the introduction, results, discussion, conclusions and abstract, that also the impact of housing changes on the rumen bacterial community was investigated.

  1. The determination of timepoint 1 does not seem to be relevant for this experiment.

Response: You are right, TP1 is not relevant to investigate the impact of DGP on the rumen bacterial community. However, the determination of the rumen bacterial community during TP1 gives a picture of the current status prior to the start of feeding the control diet. Most of the changes happened just after the change of housing and the change of diet to the control diet. Therefore, we decided to keep TP1 in the evaluation of the experiment. For the inclusion of TP1 in the paper, we added information to the introduction, results, discussion, conclusions and abstract, that also the impact of housing changes on the rumen bacterial community was investigated.

  1. The word “Scheme” should be “Figure”. (Line 79)

Response: Thank you for your attention to details.

Discussion

  1. Appropriate paragraphing is suggested.

Response: paragraphing was applicate in the introduction.

  1. The effect of different concentration of DGP should be discussed.

Response: Thank you for your comment. A comparison of the composition of the rumen bacterial community when fed diets with different levels of DGP is presented in the Results section.  It is difficult to find papers in which the addition of dried grape pomace has been tested. The main biologically active substances in grape pomace include polyphenols. Therefore, in the discussion, the results are compared to the results of papers in which the addition of polyphenols was studied.

Conclusions

  1. The conclusion is confusing and inconsistent with the majority of the discussion. If the aim of the study is to figure out the effect of grape pomace, this part should include a summary about that.

Response: Thank you for comment. The conclusions have been modified. They are now in agreement with the results of the experiment, including also indicating the effect of changing housing and feeding on the rumen bacterial community.
